# The Catalytic Role of Superparamagnetic Iron Oxide Nanoparticles as a Support Material for TiO_2_ and ZnO on Chlorpyrifos Photodegradation in an Aqueous Solution

**DOI:** 10.3390/nano14030299

**Published:** 2024-02-01

**Authors:** Wence Herrera, Joelis Vera, Edward Hermosilla, Marcela Diaz, Gonzalo R. Tortella, Roberta Albino Dos Reis, Amedea B. Seabra, María Cristina Diez, Olga Rubilar

**Affiliations:** 1Programa de Doctorado en Ciencias de Recursos Naturales, Universidad de La Frontera, Temuco 4780000, Chile; 2Programa de Doctorado en Ciencias de la Ingeniería Mención Bioprocesos, Universidad de la Frontera, Temuco 4780000, Chile; j.vera12@ufromail.cl; 3Centro de Excelencia en Investigación Biotecnológica Aplicada al Medio Ambiente CIBAMA-BIOREN, Universidad de La Frontera, Temuco 4780000, Chile; edward.hermosilla@ufrontera.cl (E.H.); marcela.diaz@ufrontera.cl (M.D.); gonzalo.tortella@ufrontera.cl (G.R.T.); cristina.diez@ufrontera.cl (M.C.D.); 4Center for Natural and Human Sciences, Universidade Federal do ABC, Santo André 09210-580, SP, Brazil; roberta.reis@ufabc.edu.br (R.A.D.R.); amedea.seabra@ufabc.edu.br (A.B.S.); 5Departamento de Ingeniería Química, Universidad de La Frontera, Temuco 4780000, Chile

**Keywords:** degradation, nanocomposites, pesticides, TCP, titanium dioxide, zinc oxide

## Abstract

Chlorpyrifos (CP) is a globally used pesticide with acute toxicity. This work studied the photocatalytic degradation of CP using TiO_2_, ZnO nanoparticles, and nanocomposites of TiO_2_ and ZnO supported on SPIONs (SPION@SiO_2_@TiO_2_ and SPION@SiO_2_@ZnO). The nanocomposites were synthesized by multi-step incipient wetness impregnation. The effects of the initial pH, catalyst type, and dose were evaluated. The nanocomposites of SPION@SiO_2_@TiO_2_ and SPION@SiO_2_@ZnO showed higher CP photodegradation levels than free nanoparticles, reaching 95.6% and 82.3%, respectively, at pH 7. The findings indicate that iron oxide, as a support material for TiO_2_ and ZnO, extended absorption edges and delayed the electron–hole recombination of the nanocomposites, improving their photocatalytic efficiency. At the same time, these nanocomposites, especially SPION@SiO_2_@TiO_2_, showed efficient degradation of 3,5,6-trichloropyridinol (TCP), one of the final metabolites of CP. The stability and reuse of this nanocomposite were also evaluated, with 74.6% efficiency found after six cycles. Therefore, this nanomaterial represents an eco-friendly, reusable, and effective alternative for the degradation of chlorpyrifos in wastewater treatment.

## 1. Introduction

Imagining modern agriculture without the use of pesticides is difficult due to the crop protection role they play [1,2]. It has been estimated that crop losses due to pests without the use of pesticides would be approximately 40% (equivalent to 2.5 trillion USD/year) [3]. However, excessive use and improper management can lead to soil and groundwater contamination [4,5,6] because pesticide residues can persist for variable periods in these environments [7]. Synthetic organophosphates (OPs) are the most broadly utilized insecticides. Chlorpyrifos (CP) [O,O-diethyl O(3,5,6-trichloro-2-pyridyl) phosphonothioate] is a broad-spectrum OP insecticide broadly used worldwide [8,9]. Low solubility in water (1.4 mg L^−1^) has been reported for CP; meanwhile, its affinity with soil organic carbon (Koc = 8.5 L g^−1^) is high. Both factors from CP have been frequently observed in rivers via surface runoff from agricultural lands and urban streams [8,10,11].

CP and its primary metabolites, chlorpyrifos oxon (CP Oxon) and 3,5,6-trichloropyridinol (TCP), cause considerable harm to the environment due to high toxicity to aquatic invertebrates and fishes, with human wellbeing concerns as well [10]. This is why it is fundamental to develop novel processes for the management and treatment of this kind of pollution. Several alternatives have been proposed to reduce, eliminate, isolate, or stabilize pesticide pollutants; among these, physical (adsorption), chemical (ozonation, UV radiation, Fenton oxidation, electro-oxidation, electro-coagulation, and advanced oxidation processes), and biological (bioremediation, phytoremediation, and bioaugmentation) treatments can be found [12]. A few years ago, using semiconductor nanoparticles (NPs) as a photocatalyst to transform many pollutants (inorganic and organic) in water became a viable water treatment technology [13]. In this context, metal nanoparticles such as TiO_2_-NP and ZnO-NP have attracted considerable interest because they are involved in advanced oxidation processes (AOP), which allows them to absorb light. Due to their photochemical properties, in addition to their ability to be a constant source of reactive oxygen species (ROS) [14], they have been successfully used to degrade pesticides such as CP, DDT, atrazine, diazinon, and malathion [15,16,17,18,19,20,21,22].

ZnO and TiO_2_, as a photocatalyst, pose a problem: the easy recombination of electron–hole pairs formed during photocatalysis, which is a limitation on the photocatalytic performance of these nanoparticles [23,24]. Nevertheless, one of the principal characteristics of nanomaterials is that they can be functionalized, or other compounds (organic or inorganic) can be chemically attached to their surface to enhance their affinity towards a target [25,26]. Recently, a wide range of chemical methods have been reported to synthesize functionalized NPs [27]. Currently, the most used functionalized nanoparticles to eliminate environmental pollutants are bimetallic nanoparticles, which have been successfully tested [28,29]. Regarding pesticide degradation, bimetallic nanoparticles have shown high degradation rates, even higher than those exhibited by non-functionalized metal nanoparticles. The addition of some metals to create nanocomposites enhanced the reactivity of photocatalytic NPs due to the metals acting as an electron sink/trap on the semiconductor surface, abating the electron–hole recombination process. The activation energy of the reaction increases the interaction between compounds and the degradation rate.

Functionalized NPs based on magnetic nanoparticles are widely used because they are a type of NP that can be followed, utilized, and targeted via an external magnetic field [30]. In this context, superparamagnetic iron oxide nanoparticles (SPIONs) are considered to have high potential to be applied in pesticide degradation due to their easy manipulation on the field. However, the effect of functionalized SPIONs on CP degradation is not clear; this research aimed to evaluate the influence of SPIONs as a support material of TiO_2_ and ZnO on the chlorpyrifos degradation rate in an aqueous medium.

## 2. Materials and Methods

### 2.1. Synthesis of Fe_3_O_4_-NP

Magnetite nanoparticles were prepared via the chemical co-precipitation method described by Jesus et al. [31]. A total of 0.5 g of FeCl_3_•6H_2_O and 0.2 g of FeCl_2_•4H_2_O were dissolved in 5 mL of HCl (0.25 M) and stirred at room temperature for 15 min. Afterwards, 50 mL of an NH_4_OH solution (0.7 M) was added dropwise under vigorous stirring. The final product was separated by a super magnet and washed five times with ethanol and water. Then, the nanoparticles were dried in a desiccator. The obtained nanoparticles were characterized via transmission electronic microscopy (TEM, Libra 120 Zeiss, Zeiss group, Oberkochen, Germany, X-ray diffraction (XRD, Rigaku Smartlab, Rigaku Corporation, Tokyo, Japan), and Fourier-transform infrared spectroscopy (FTIR, Agilent Technologies, Santa Clara, CA, USA).

### 2.2. Synthesis of SPION@SiO_2_

A silicon dioxide layer (SiO_2_) is one of the most used materials in the functionalization of magnetic nanoparticles because it can provide stability and dispersion. Additionally, the silicon dioxide layer can be combined with other materials, such as TiO_2_ and ZnO nanoparticles [32,33,34]. Chi et al. used a modified Stöber method to prepare the SiO_2_ interlayer [35]. Prepared Fe_3_O_4_ NP (0.4 g) was added to a mixture of ammonia aqueous solution (4.8 mL), deionized water (40 mL), and ethanol (160 mL) and dispersed by an ultrasonication bath for 1 h. Then, 1.4 mL of tetraethyl orthosilicate (TEOS) was added drop by drop and stirred overnight. Afterward, NPs were separated by a neodymium magnet, and thorough washing with ethanol and water was carried out. Finally, nanoparticles were dried at 70 °C. The obtained product was characterized by TEM, FTIR, and XRD techniques. 

### 2.3. Synthesis of SPION@SiO_2_@TiO_2_ and SPION@SiO_2_@ZnO Nanocomposites

The synthesis of bimetallic nanoparticles was performed by a multi-step incipient wetness impregnation method. SPION@SiO_2_ nanoparticles were suspended in water and precursor salts (Zn (NO_3_)_2_ to obtain ZnO and titanium isopropoxide (TTIP) to obtain TiO_2_), and dispersed using an ultrasonic bath for 30 min. To prepare SPIONs@SiO_2_@TiO_2_, 9 mL of concentrated acetic acid was added and stirred continuously for 3 h at 90 °C. Subsequently, the solution was permitted to cool to room temperature. Then, 10 mL of NaOH (7 M) was added to precipitate a TiO_2_ shell over the SPION@SiO_2_. The resulting nanocomposite was separated using a neodymium magnet, washed with deionized water and ethanol, and dried at 60 °C. To prepare SPIONs@SiO_2_@ZnO, the solution was heated to 80 °C, and 50 mL of NaOH (0.2 M) was added drop by drop. Next, the solution was mixed vigorously for two hours, and the resulting nanocomposite was separated using a neodymium magnet and washed with deionized water and ethanol. Finally, the product was calcined in a furnace at 400 °C for 3 h. The obtained product was characterized by the TEM, FTIR, and XRD techniques. The specific surface areas were calculated using the Brunauer–Emmett–Teller (BET) theory, and the pore size distributions and pore volumes were calculated with desorption data from adsorption–desorption isotherms (based on the Barrett–Joyner–Halenda (BJH) theory) using a Quantachrome Nova 1000e (Anton Paar QuantaTec, Golf, FL, USA) surface area and pore size analyzer.

### 2.4. Photocatalytic Degradation

The assays of photocatalytic degradation of CP were performed in 100 mL Erlenmeyer flasks containing 50 mL of CP (50 mg·L^−1^, 1:1 methanol: water) and 0.1 G·mL^−1^ of each catalyst synthesized previously (SPION, ZnO, TiO_2_, SPION@SiO_2_, SPION@SiO_2_@TiO_2_ and of SPION@SiO_2_@ZnO), under 120 rpm stirring at ≈25 °C (Figure 1). The photoreactor was equipped with three projection lamps (TLK 40 W/10 R 25PK) with a main emission wavelength of 370 nm. To obtain adequate dispersion of the catalyst, the solutions were sonicated for 10 min in dark conditions before testing started. One-milliliter samples were taken every 24 h for 120 h, and the magnetic nanoparticles and nanocomposites were separated using a neodymium magnet and filtered (PTFE Syringe Filter, Agilent Technologies, Waldbronn, Germany, 13 mm, 0.22 µm). The concentration of CP in the supernatant was determined by high-performance liquid chromatography (HPLC Merck Hitachi L-2130, Hitachi, Tokyo, Japan). Separation was achieved using a C_18_ column (250 × 4.6 mm, particle size 5 µm). In the assays using TiO_2_ and ZnO NPs as photocatalysts, the aliquots were centrifugated at 10,000 rpm × 10 min to separate nanoparticles from the solution. CP degradation efficiency was calculated using Equation (1):(1)$$\mathrm{CP}\;\mathrm{degradation}\;\left(\%\right)=\left(\frac{C_{0}-C_{t}}{C_{0}}\right)\times 100$$
where *C_o_* (ppm) is the initial chlorpyrifos concentration, and *C_t_* (ppm) is the concentration of CP at a specific time. 

### 2.5. Kinetic Analyses

The degradation of CP from liquid medium was fitted to a first-order model following Equation (2):(2)$$ln\frac{C_{t}}{C_{0}}=e^{-kt}$$
where *C*_0_ is the initial CP concentration in the aqueous solution, *C_t_* is CP concentration at time *t*, *k* is the degradation rate constant (h^−1^), and *t* is the reaction time (h). Degradation half-life (*T*_1/2_) is the time in which CP loses 50% of the original concentration, and it was calculated according to Equation (3):(3)T12=ln2k

### 2.6. Statistical Analyses

The trials were replicated three times under each set of conditions. Subsequently, the data analysis utilized either a one-way analysis of variance (ANOVA) or the Kruskal–Wallis test. In cases where statistical variances were identified, differences between means were determined through Tukey’s minimum significant difference test (*p* < 0.05). For the statistical analysis, we utilized IBM SPSS Statistics (IBM Corp. Released 2021, Version 28.0, Armonk, NY, USA: IBM Corp).

### 2.7. Proposed Degradation Pathway of Chlorpyrifos by SPION@SiO_2_@TiO_2_

Samples (1 mL) were taken for each flask and filtered through a PTFE 0.22 µm filter before analysis. The degradation pathway of chlorpyrifos on SPION@SiO_2_@TiO_2_ was studied with HPLC MS/MS (Sciex 3200 Q-trap, Sciex, MA, USA) using a C18 column (Chromolit RP-8e, 4.6 µm × 100 mm, Merck, Darmstadt, Germany). The mobile phase was 80% acetonitrile injected at a flow rate of 0.2 mL·min^−1^ for 15 min.

## 3. Results and Discussion

### 3.1. Characteristics of Synthesized Nanoparticles

#### 3.1.1. XRD Analysis

The XRD patterns of SPION, ZnO, TiO_2_, SPION@SiO_2_, SPION@SiO_2_@TiO_2_, and SPION@SiO_2_@ZnO nanoparticles are shown in Figure 2. As can be observed, the XRD analysis showed magnetite (Fe_3_O_4_) as the main crystalline phase in all samples (JCPDS: 65-3107), except for ZnO and TiO_2_ NP (Figure 2B,C), indicating that the structure of Fe_3_O_4_ remained stable after the functionalization process. All broad diffraction peaks of (220), (311), and (511) and at around 2θ of 30.4°, 36.4°, and 57.4° suggested that the crystal size of Fe_3_O_4_ particles was quite small. The high background of the XRD patterns can be attributed to the low crystallization of particles. Figure 2B shows the XRD of ZnO NP, which has all the spectral peaks indexed to ZnO with hexagonal wurtzite (JCPDS data file No. 36–1451). Meanwhile, Figure 2C shows a well-crystallized anatase profile for TiO_2_ NPs (JCPDS data file No. 21-1272). When comparing ZnO picks in Figure 2C,E, no displacement is observed; therefore, the formation of heterostructures cannot be confirmed. The same result was obtained from comparing TiO_2_ picks in Figure 2C,F.

As shown in Figure 2D–F, no peak corresponding to SiO_2_ was found because of the amorphous structure of the synthesized SiO_2_ particles [36,37]. SPION@SiO_2_@TiO_2_ (Figure 2F) present previously described peaks that have similar characteristics, which indicates that Fe_3_O_4_ is present in the nanocomposite. In addition, a new diffraction peak appeared at 25.4, corresponding to the (101) crystal plane of anatase (TiO_2_) (JCPDS: 21-1272). In Figure 2E, no distinctive diffraction peaks of ZnO were identified, possibly due to the relatively small quantity of ZnO, resulting in subtle or undetectable signals during the analysis. To confirm the presence of TiO_2_ and ZnO in nanocomposites, energy-dispersive X-ray spectroscopy was performed (Appendix A).

#### 3.1.2. FT-IR Analysis

Figure 3 shows the FTIR spectra of SPION, ZnO, TiO_2_, SPION@SiO_2_, SPION@SiO_2_@TiO_2_, and SPION@SiO_2_@ZnO composites. All the spectra evidenced a broad absorption peak located in the fingerprint region of 800 to 500 cm^−1^, which corresponded to a metal–oxygen linkage like in Fe-O, ZnO, and Ti-O. The characteristic bands of silica are visible at ≈800 and ≈1050 cm^−1^ due to Si–O–Si symmetric stretching and Si–O–Si asymmetric vibration, respectively (Figure 3D–F). The intensity of the absorption band at 1050 cm^−1^ (Figure 3E) decreases (in comparison with Figure 3D), suggesting that SPION@SiO_2_ was covered with TiO_2_. Samples E–F presented a large band at 3300 cm^−1^ due to the -OH stretch, probably as a result of the presence of water or ethanol in the sample.

#### 3.1.3. TEM Analysis

The synthesized uncoated SPION showed a predominantly spherical shape (Figure 4A) and a normal size distribution with a mean TEM size of 11.95 nm (Figure 4E). The synthetized SPION presented superparamagnetic behavior since no coercivity or remanence were observed (Appendix A) with a maximum saturation at 60 emu g^−1^. This saturation is higher than that reported in the literature (52.3 emu g^−1^) for 7 nm SPION [38]. Figure 4B shows the SPION coated with SiO_2_ (SPION@SiO_2_), which has a typical core–shell structure. The average thickness of the SiO_2_ layer was 5.21 nm, increasing the average size of the nanocomposite to 17.57 nm (Figure 4F). SPION@SiO_2_ was used as support of TiO_2_ NP, which adhered to the surface of the nanocomposite (Figure 4C), forming SPION@SiO_2_@TiO_2_ whose average size was 17.12 nm (Figure 4G). Similarly, SPION@SiO_2_@ZnO nanocomposites were synthesized, maintaining a spherical shape (Figure 4D) with an average size of 18.84 nm (Figure 4H). 

#### 3.1.4. Porosity and Surface Area of SPION@SiO_2_@TiO_2_ and SPION@SiO_2_@ZnO Nanocomposites

The N_2_ adsorption/desorption for synthetized nanocomposites is shown in Appendix A. SPiON@SiO_2_@TiO_2_ and SPiON@SiO_2_@ZnO showed a type IV hysteresis loop according to the IUPAC classification [39]. This type is mesoporous, which can be corroborated by having a pore diameter between 2 and 50 nm (Table 1). In general, it can be observed that SPiON@SiO_2_@TiO_2_ presented a higher surface area (243.22 m^2^·g^−1^), pore diameter (3.84 nm), and pore volume (0.19 cm^3^·g^−1^) than SPiON@SiO2@ZnO, which presented a surface area of 108.79 m^2^·g^−1^, pore diameter of 3.27 nm, and pore volume of 0.13 cm^3^·g^−1^.

Nanoparticle surface area and pore size are linked to the photocatalytic potential of the nanocomposites since the adsorption of the reactive species is among the crucial rate-determining steps of the catalytic reaction [40]. It has been reported that the use of ZnAl_2_O_4_ nanoparticles as the photocatalyst in Red 141 dye degradation is more effective when a larger pore size is employed [41]. A larger surface area will have more active sites for ROS production, leading to a higher degradation rate [42].

### 3.2. Photocatalytic Degradation

#### 3.2.1. Initial pH Effect

The pH effect on the photodegradation of CP is shown in Figure 5. For both nanocomposites SPIONS@TiO_2_ and SPIONS@SiO_2_@ZnO, a correlated effect is observed in which CP degradation increases, which increases the initial pH (Figure 5A,B). In the case of SPIONS@TiO_2_ nanoparticles (Figure 5A), the best photodegradation rate was 94% at a pH of 9. On the other hand, SPIONS@ SiO_2_@ZnO nanoparticles (Figure 5B) showed slightly less degradation than SPIONS@TiO_2_ (91.3% at pH 9). The influence of pH on photocatalytic reactions is frequently ascribed to the surface charge of the catalyst [23]. The point of zero charge (pzc) for TiO_2_ and ZnO is 6.5 and 9, respectively [43,44]. At a pH below pzc, the catalyst surface carries a positive charge, whereas at a higher pH than pzc, it becomes negatively charged. This results in electrostatic attraction or repulsion between the catalyst and the target compound, contingent on the ionic form of the compound. Consequently, this either enhances or inhibits the photodegradation efficiency, respectively. 

The major causes of degradation enhancement are as follows: (i) when there is a large amount of OH-, the catalyst surface will be negatively charged at higher pH values, causing a high concentration of OH-, which will progressively yield oxidative ^●^OH species; (ii) the electrostatic attraction between the CP molecules and TiO_2_ or ZnO is greater at an alkaline pH, so CP molecules are allowed to reach the catalyst’s surface easily. Therefore, they are more favorable for the efficient generation of ^●^OH and O^2−●^in an alkaline medium and achieve higher photodegradation efficiency.

Similar results were found by Fadei and Kargar [45], who evaluated the degradation of chlorpyrifos at pH 5, 7, and 9. They found that both TiO_2_ and ZnO NP have better degradation rates at an alkaline pH than at an acidic pH. On the other hand, at pH 11, there was a decrease in photocatalysis for both nanomaterials, which could be due to the pH. A large amount of OH ions can screen the charges and reduce the movement of reactive oxygen species responsible for the transformation of CP molecules.

#### 3.2.2. Nanoparticle Dose Effect

To examine the effect of SPION@SiO_2_@TiO_2_ and SPION@SiO_2_@ZnO concentration on photodegradation, the initial catalyst amount was systematically varied in the range of 0.012–0.2 g (see Figure 6). The degradation of CP can be seen in Figure 6; after 120 h of continuous irradiation, chlorpyrifos degradation was 92%, 91.3%, 78.6%, 58%, and 28.7% with a SPION@SiO_2_@TiO_2_ NP amount of 0.2, 0.1, 0.05, 0.25, and 0.12 g. L^−1^, respectively (Figure 6A). Meanwhile, the degradation rate was 87.3%, 86.7%, 72.6%, 50%, and 25.3% with SPION@SiO_2_@ZnO amounts of 0.2, 0.1, 0.05, 0.25, and 0.12 g·L^−1^, respectively (Figure 6B). In both cases, no significant differences between 0.2 and 0.1 g·mL^−1^ were found.

#### 3.2.3. Effect of Catalyst Type on CP and TCP Degradation

Figure 7 shows the effect of the type of nanomaterial on CP degradation. Control treatments with SPION and SPION@SiO_2_ materials showed the lowest degradation rates (3%), which had no significant differences compared with the control without nanoparticles (3.3%). When free nanoparticles of ZnO and TiO_2_ were used, the CP photodegradation increased to 45.2% and 52.3%, respectively, after 120 h. In the assays with the nanocomposites, SPION@SiO_2_@ZnO and SPION@SiO_2_@TiO_2_ reached 82.3% and 95.7% of CP degradation after 120 h—1.82-fold higher than the free nanoparticle photocatalysts. These findings suggest a synergistic effect between SPION@SiO_2_ and ZnO or TiO_2_, enhancing their photocatalytic activity for CP degradation when combined.

Both semiconductor nanoparticles (TiO_2_ and ZnO) exhibited similar photocatalytic activity, because both are good producers of reactive oxygen species (ROS). However, it has been reported that TiO_2_ NP generated more ROS than ZnO NP under the same conditions [46], mainly due to the vulnerability of ZnO NP to photocorrosion, leading to a decrease in ZnO photocatalytic activity in aqueous solutions [46,47]. TiO_2_ has been reported to be an efficient and viable photocatalyst for the degradation and mineralization of several organic pollutants in the presence of UV.

On the other hand, it is important to mention that the functionalization of semiconductor nanoparticles (ZnO and TiO_2_) with SPIONS has been shown to encourage the degradation of CP, mainly because the presence of Fe_3_O_4_ extended the absorption edges of the nanocomposite SPION@TiO_2_, or SPION@ZnO postponed the electron–hole recombination [48]. Similar results were found by Elshypany et al. [49], who evaluated the photodegradation of the dye methylene blue in the presence of ZnO and the nanocomposite SPION@ZnO and reported that the degradation rate increased when ZnO was functionalized with Fe_3_O_4_. Likewise, Fadei and Kargar [45] reported a much faster degradation (30 min) of CP (6 mg L^−1^) using ZnO and TiO_2_ NP compared to ours. It has been reported that, in photocatalytic assays, as the pesticide concentration increases, the degradation rate decreases. This can be attributed to pesticides binding to active sites on nanocatalysts, as well as a reduction in UV light penetration into the solution. Consequently, both effects lead to a reduction in reactive oxygen species (ROS) production [50]. This can explain the differences between our results and those obtained by Fadei and Kargar, who used 6 mg·L^−1^ CP in the assays.

Finally, it is important to mention that the degradation of chlorpyrifos leads to the formation of two metabolites, (1) chlorpyrifos-oxon and (2) 3,5,6-trichloro-2-pyrinidol (TCP). The latter is highly toxic due to the presence of chlorine atoms in its structure, and it has higher water solubility (80.9 mg L^−1^) than CP (1.05 mgL^−1^) [51]. Figure 8 shows the effect of different nanoparticles on TCP degradation. It can be observed that, initially, the TCP concentration is close to 0 because chlorpyrifos has not been degraded in the medium; however, it increases as the degradation of chlorpyrifos in the medium increases, with the highest concentration being reached at 96 h using SPIONS@SiO_2_@TiO_2_ (5.93 mg. L^−1^). It can also be observed that, for all cases, except for the control and magnetite (Fe_3_O_4_), there is a decrease in the concentration of TCP in the medium, which indicates that there is a photodegradation of this metabolite in the aqueous medium. SPIONS@SiO_2_@TiO_2_ showed the highest degradation rate (56.27%). Žabar et al. [52] reported a degradation of 5.9% of the initial TCP concentration (52 mg·L^−1^) after 120 min of treatment in a photoreactor with UV light and TiO_2_ nanoparticles.

It has been reported that functionalized ZnO and TiO_2_ mainly produce •OH and O_2_^•−^, which act together in the degradation process [53,54]. In general, SPION@SiO_2_@ZnO and SPION@SiO_2_@TiO_2_ are very similar; however, the difference in their degradation rate stems from the fact that SPION@SiO_2_@TiO_2_ has a higher surface area, which improves the ability to harvest incident light and create more surface-active sites [55].

#### 3.2.4. Kinetic Study

To investigate the kinetics of CP degradation, the experiment was conducted under optimized conditions. For both SPION@SiO_2_@TiO_2_ and SPION@SiO_2_@ZnO, the reaction followed a pseudo-first-order kinetic (Figure 9A and Figure 9B, respectively), with good correlation with the linear regression coefficient (R^2^ = 0.9189 and R^2^ = 0.9359). The degradation rate constant (*k*) was almost twice as high when using SPION@SiO_2_@TiO_2_ (0.025 h^−1^) than SPION@SiO_2_@ZnO (0.014 h^−1^). Meanwhile, the half-life was 27.5 h for SPION@SiO_2_@TiO_2_ and 48.14 h for SPION@SiO_2_@ZnO. These times are considerably shorter than those reported for hydrolysis in water and aqueous solutions (56 days and 156 h, respectively) [56,57]. Likewise, times are longer than those reported by Budarz et al. [10], who found a T^1^/_2_ of 443 min in CP degradation (20 mg·L^−1^) using TiO_2_ in a bioreactor. Meanwhile, for ZnO nanoparticles degrading CP (10 mg·L^−1^), the T^1^/_2_ has been reported as 58.23 min [58].

#### 3.2.5. Reuse and Stability of Photocatalysts

SPION@SiO_2_@TiO_2_ exhibited the highest degradation rates for CP. Consequently, it was chosen for the evaluation of stability, recovery, and reusability in six consecutive CP degradation runs. Upon completion of the CP degradation reaction, the photocatalyst was separated from the solution using an external magnetic field, subjected to a wash step, and reused. Figure 10 shows the CP degradation by SPIONS@SiO_2_@TiO_2_ in six sequential cycles. We observed a decrease after each cycle, and 74.67% of CP degradation was observed after six uses. This value was statistically different compared to the first use (92.63%). These results are similar to those reported by Farahbakhsh et al. [59], who found a decrease in the degradation rate after seven uses (81%), with the first use (98%) using biochar/CdS-Fe_3_O_4_ as a photocatalyst. This decrease in degradation rate may be due to a loss of photocatalytic nanomaterial in the washes, a decrease in the TiO_2_ layer, the oxidation of the nanoparticles, or agglomeration of the magnetic nanoparticles.

After use, the nature of the nanocomposites was evaluated with FTIR to determine any change in the chemical properties and possible adsorption of CP on the catalyst surface. Figure 11 shows no difference between unused SPION@SiO_2_@TiO_2_ and SPION@SiO_2_@TiO_2_ after their use in photocatalysis assays.

#### 3.2.6. Proposed Degradation Pathway

The degradation pathway of chlorpyrifos on SPION@SiO_2_@TiO_2_ was studied using HPLC MS/MS (Sciex 3200 Q-trap). The results revealed seven intermediate products (Figure 12). These results were similar to those reported by Teymourinia et al. [50], who described the process as follows: the CP molecule that reacted with the OH radical photogenerated and oxidized the P=S bond, yielding chlorpyrifos oxon. At the same time, CP underwent oxidation and ring hydroxylation/dechloration, leading to intermediate P1. Both chlorpyrifos oxon and P1 formed diethyl hydrogen phosphate by hydrolysis. Then, chlorpyrifos oxon lost two molecules of ethanol and a phosphate group, producing 3,5,6-trichloro pyridinol by hydrolysis. This molecule underwent dehydroxylation and dechlorination to produce mono- and dichlorinated pyridines. The dechlorination of chlorinated pyridines resulted in the formation of pyridine. Finally, the organic species were decomposed to NHO_4_+ CO_2_ + H_2_O.

## 4. Conclusions

In this study, the role of various operational parameters and their effect on the degradation of CP was evaluated. The results confirm that SPIONS play an important role in the delay of electron–hole recombination and enhance the photodegradation of CP when it is supporting ZnO and TiO_2_ nanoparticles. Both nanocomposites exhibited optimal CP degradation rates at pH 9; however, no significant differences were observed at pH 7. This suggests that these materials can work efficiently at a pH range close to neutral. The SPION@SiO_2_@TiO_2_ photocatalyst showed a high potential to degrade the TCP formed during the CP degradation, making it an alternative for use in systems with both pollutants. Finally, we demonstrate that SPION@SiO_2_@TiO_2_ can be reused in at least six runs. 

## Figures and Tables

**Figure 1 nanomaterials-14-00299-f001:**
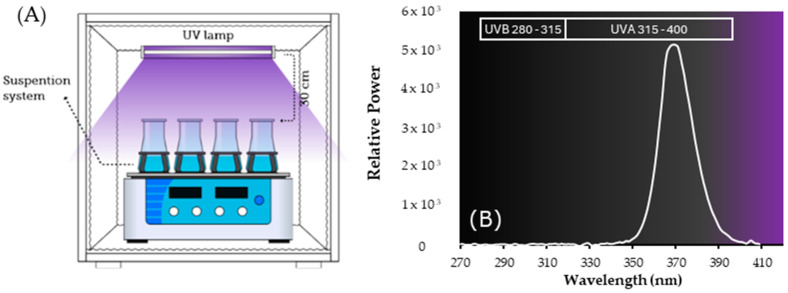
(**A**) Reaction system for photocatalytic degradation studies. (**B**) Emission spectrum of 40 W Philips lamp (λmax = 370 nm).

**Figure 2 nanomaterials-14-00299-f002:**
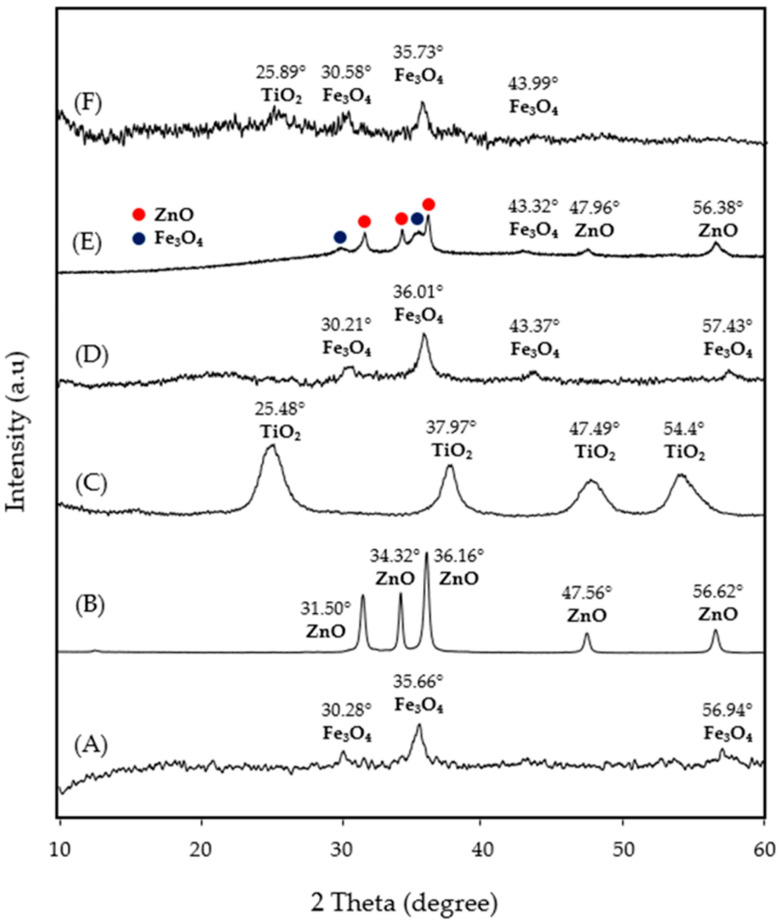
XRD patterns of (**A**) SPION, (**B**) ZnO, (**C**) TiO_2_, (**D**) SPION@SiO_2_, (**E**) SPION@SiO_2_@ZnO, and (**F**) SPION@SiO_2_@TiO_2_ nanoparticles.

**Figure 3 nanomaterials-14-00299-f003:**
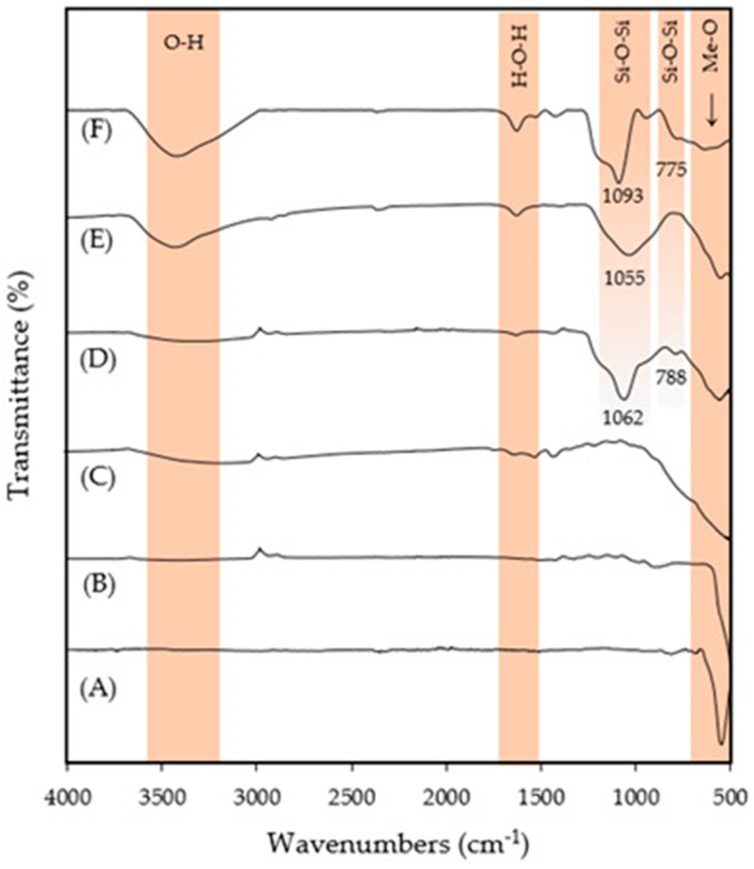
FTIR spectra of (**A**) SPION, (**B**) ZnO, (**C**) TiO_2_, (**D**) SPION@SiO_2_, (**E**) SPION@SiO_2_@TiO_2_, and (**F**) SPION@SiO_2_@ZnO nanoparticles.

**Figure 4 nanomaterials-14-00299-f004:**
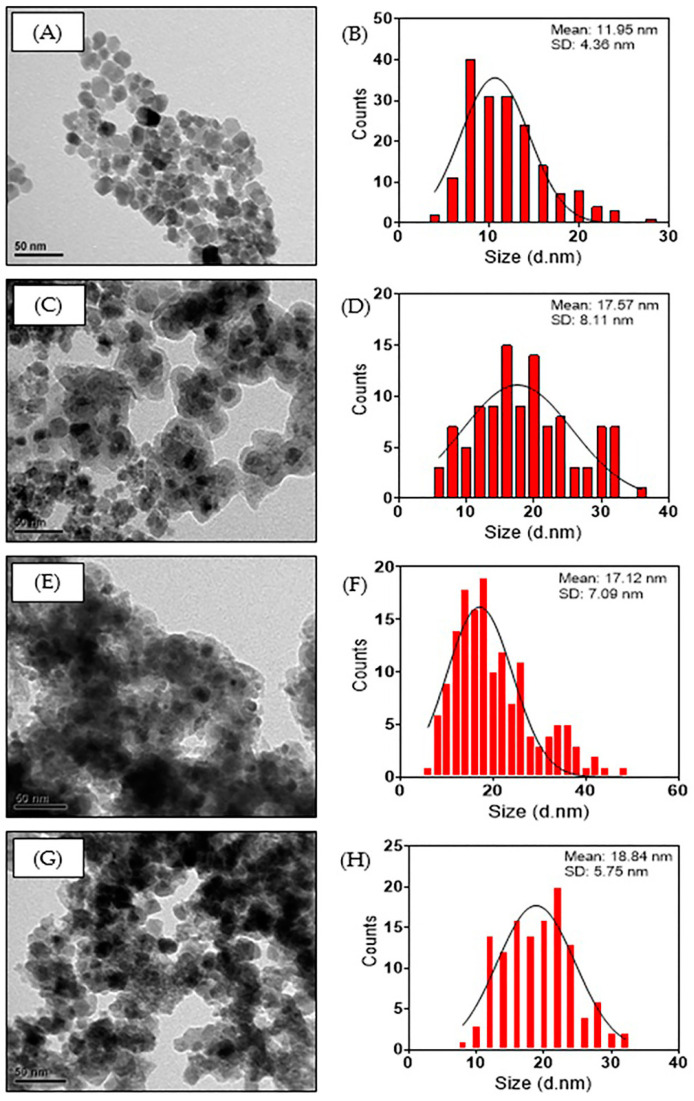
TEM images and size distribution of (**A**,**B**) SPION, (**C**,**D**) SPION@SiO_2_, (**E**,**F**) SPION@SiO_2_@TiO_2_, and (**G**,**H**) SPION@SiO_2_@ZnO nanoparticles.

**Figure 5 nanomaterials-14-00299-f005:**
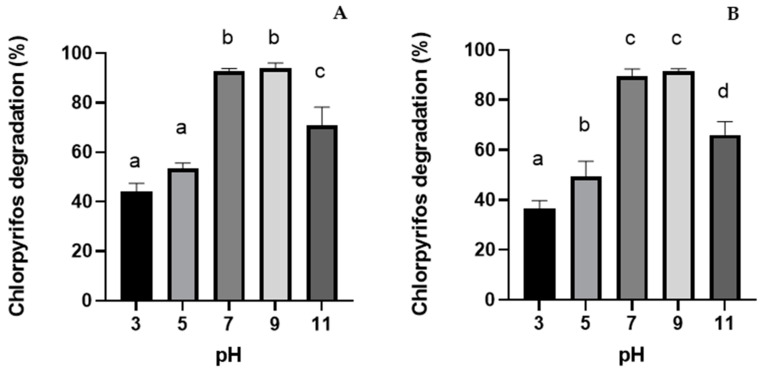
Chlorpyrifos degradation rate at different pH values using (**A**) SPION@TiO_2_ NP and (**B**) spion@ZnO NP. (Assay conditions: 50 mg·L^−1^ CP, 0.1 g·L^−1^ nanomaterial dose, 120 rpm, 25 °C, and UV-A irradiation.) Letters indicate statistical differences (*p* < 0.05).

**Figure 6 nanomaterials-14-00299-f006:**
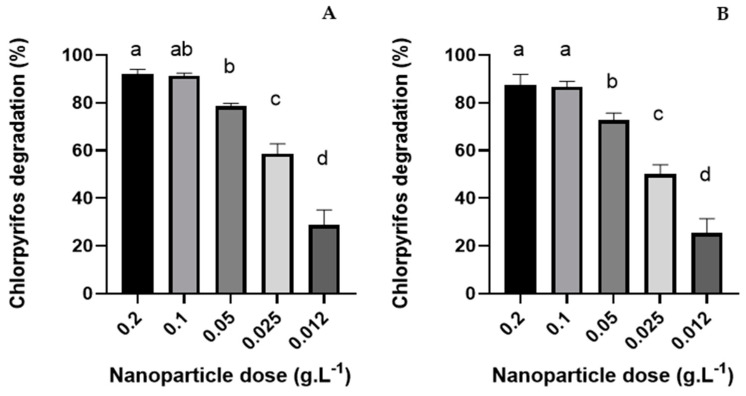
Chlorpyrifos degradation rate at different nanoparticle doses using (**A**) SPION@SiO_2_@TiO_2_ NP and (**B**) SPION@SiO_2_@ZnO NP. Assay conditions: 50 mg·L^−1^ CP, pH 7, 120 rpm, 25 °C, and UV-A irradiation. Letters indicate statistical differences (*p* < 0.05).

**Figure 7 nanomaterials-14-00299-f007:**
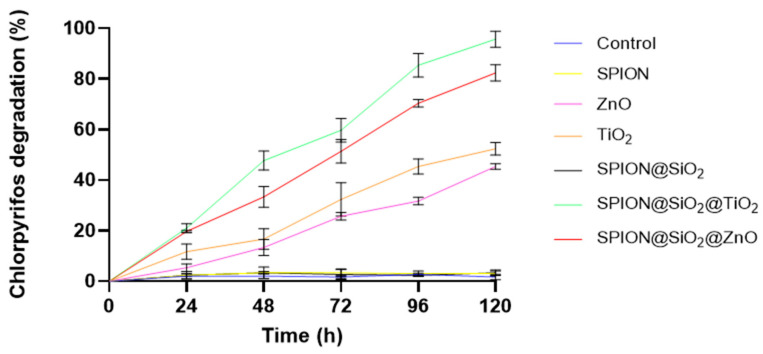
Chlorpyrifos degradation rate in the presence of different nanoparticles. Assay conditions: 50 mg·L^−1^ CP, pH 7, 1 g·mL^−1^ nanomaterial dose, 120 rpm, 25 °C, and UV-A irradiation.

**Figure 8 nanomaterials-14-00299-f008:**
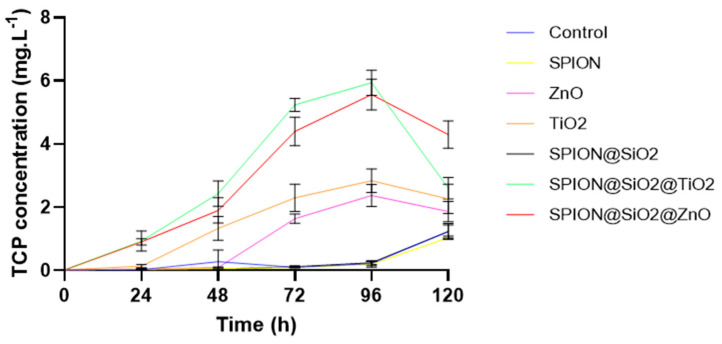
TCP (3,5,6-trichloropyridinol) concentration in the presence of different nanocatalysts. Assay conditions: 50 mg L^−1^ CP, 0.1 mg·L^−1^ nanomaterial dose, pH 7, 120 rpm, 25 °C, and UV-A irradiation.

**Figure 9 nanomaterials-14-00299-f009:**
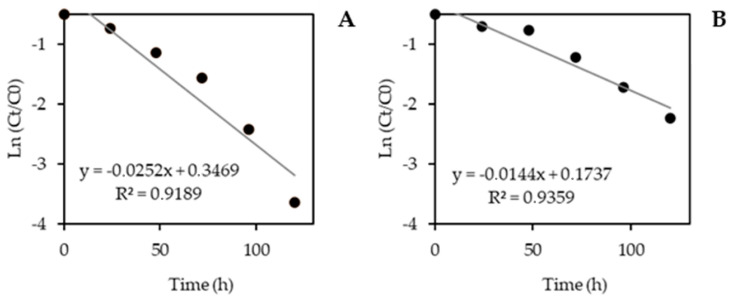
Pseudo-first-order kinetic for CP photocatalytic degradation using (**A**) SPION@SiO_2_@TiO_2_ and (**B**) SPION@SiO_2_@ZnO. Assay conditions: 50 mg L^−1^ CP, 0.1 mg·L^−1^ nanomaterial dose, pH 7, 120 rpm, 25 °C, and 120 h UV-A irradiation.

**Figure 10 nanomaterials-14-00299-f010:**
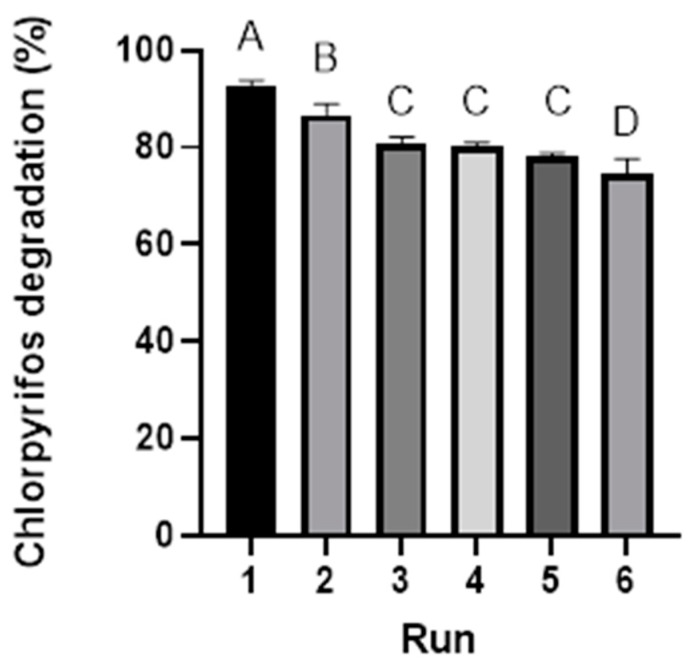
Reusability of SPION@SiO_2_@TiO_2_ nano-photocatalyst for CP degradation rate. Assay conditions: 50 mg L^−1^ CP, 0.1 mg·L^−1^ nanomaterial dose, pH 7, 120 rpm, 25 °C, and 120 h UV-A irradiation. Letters indicate statistical differences (*p* < 0.05).

**Figure 11 nanomaterials-14-00299-f011:**
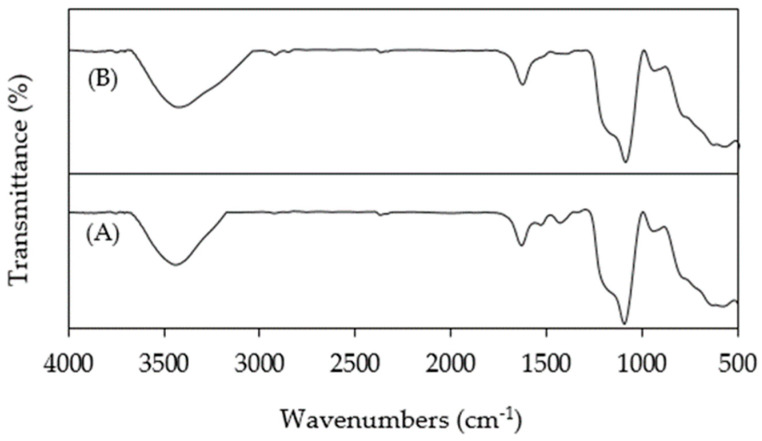
FTIR spectra of (**A**) unused SPION@SiO_2_@TiO_2_ and (**B**) used SPION@SiO_2_@TiO_2_.

**Figure 12 nanomaterials-14-00299-f012:**
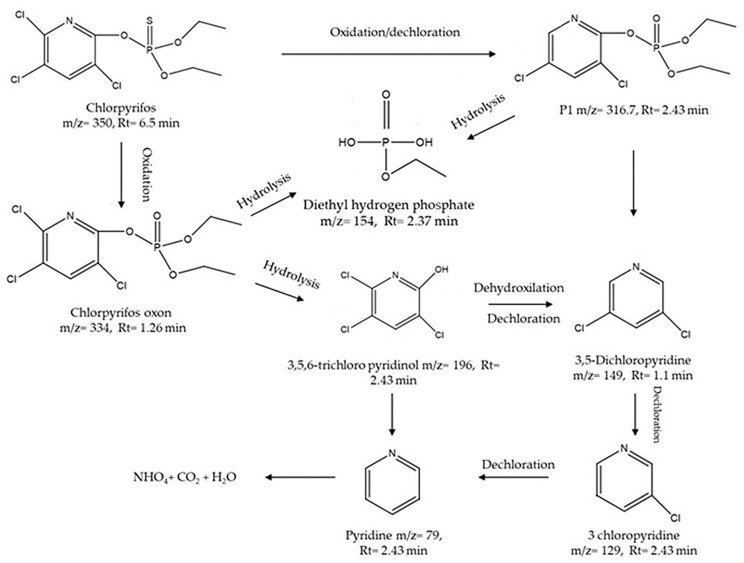
Proposed pathway of photocatalytic degradation of chlorpyrifos by SPION@SiO_2_@TiO_2_.

**Table 1 nanomaterials-14-00299-t001:** BJH and BET analysis of synthesized nanocomposites.

Sample	A_BET_ (m^2^·g^−1^)	Pore Diameter (nm)	Pore Volume (cm^3^·g^−1^)
SPiON@SiO_2_@TiO_2_	243.22	3.84	0.19
SPiON@SiO_2_@ZnO	108.79	3.27	0.13

## Data Availability

Data are available on request.

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
