# Peer review of "The Catalytic Role of Superparamagnetic Iron Oxide Nanoparticles as a Support Material for TiO2 and ZnO on Chlorpyrifos Photodegradation in an Aqueous Solution"

_nanomaterials, 2024, doi:10.3390/nano14030299_

Round 1

Reviewer 1 Report

Comments and Suggestions for Authors

The manuscript reports on the photodegradation of the chlorpyrifos in aqueous solution using nanocomposites of superparamagnetic iron oxide nanoparticles and TiO2 or ZnO nanoparticles.

The manuscript is interesting, but there are some issues that need to be carefully addressed before the manuscript can be published.

1)    My main concern regards characterization of the composites. The only technique which confirms the presence of ZnO or TiO2 is EDX microanalysis. The spectra show also other peaks at 1.5 keV in Fig. S2 for the sample Fe3O4@SiO2@TiO2 and around 3.2 keV in Fig. S3 for the sample Fe3O4@SiO2@ZnO. Are the ZnO or TiO2 species on the SPION surface or embedded within the structure? Being the entire manuscript based on the photocatalytic activity of the nanocomposites the above-mentioned points are crucial.

2)      The FT-IR spectra present some unusual features around 2900-3000 cm-1. Being a transmittance spectrum, the peak should be towards the lower part of the spectrum, as in the 500-1100 cm-1 region.

3)      Time should be reported on the X-scale of figure 9.

4)      To have a 100% degradation 120 h are needed. This is actually a too long time, and after 24 h only 20% of the chlorpyrifos was degraded. The authors should comment on results of photocatalysis with respect to literature data for this or a similar pesticide.

5)   Have the authors checked the nature of the composites after application to photodegrade the pesticide, i.e. after the 120 h?  Have been the same composite samples applied for more photodegradation tests?

6)      Please define TTIP at line 113, page 4.

Overall, the manuscript needs to be carefully revised as suggested, before it can be accepted for publication.

Comments on the Quality of English Language

English need to be corrected by an English native speaker.

Author Response

We are thankful for the reviewers’ valuable comments and contributions to the manuscript. We hope that the present improved version of the manuscript is suitable for publication in Nanomaterials.

  • My main concern regards characterization of the composites. The only technique which confirms the presence of ZnO or TiO2 is EDX microanalysis. The spectra show also other peaks at 1.5 keV in Fig. S2 for the sample Fe3O4@SiO2@TiO2 and around 3.2 keV in Fig. S3 for the sample Fe3O4@SiO2@ZnO. Are the ZnO or TiO2 species on the SPION surface or embedded within the structure? Being the entire manuscript based on the photocatalytic activity of the nanocomposites the above-mentioned points are crucial.

We conducted repeated FTIR and XRD analyses for both nanocomposites but could not definitively confirm the formation of a heterostructure due to the absence of peak displacement. Regarding the SPION@SiO2@TiO2 samples, the interference of the Fe3O4 signal makes it challenging to accurately determine any shift in the peaks. Despite efforts to mitigate noise, no discernible shift in peaks compared to ZnO was observed. However, this analysis does not conclusively establish the presence of a heterostructure. To further investigate, small-angle X-ray Scattering (SAXS) would be preferable, but unfortunately, our current XRD equipment does not support this. We acknowledge the need for this analysis in future studies. Nonetheless, TEM images present evidence countering the formation of a core-shell structure.

On the other hand, the picks present at 1.5 and 3.2 keV could correspond to Mg and K, respectively. So, we can attribute it to the composition of the sample holder on which the samples are loaded for analysis. On the other hand, the elemental mapping results (Figure S2 and Figure S3) show that the elements. Fe, Si, O, Zn, or Ti are homogeneously distributed in the nanocomposite samples.

  • The FT-IR spectra present some unusual features around 2900-3000 cm-1. Being a transmittance spectrum, the peak should be towards the lower part of the spectrum, as in the 500-1100 cm-1 region.

Samples E and F presented a broad band at 3300 cm-1, attributed to the -OH stretch, resulting from the presence of water or ethanol molecules adsorbed on the sample. The peak observed between 2900-3000 cm-1 may be attributed to some organic residue present in the equipment during the sample measurement. However, these peaks does not interfere with the characterization, as the metal bonds crucial for characterization occur in the region around 500-1000 cm-1.  (Line 230-232).

  • Time should be reported on the X-scale of figure 9. (Line 357)

The figure 9 was corrected in the revised manuscript. (Line 408)

  • To have a 100% degradation 120 h are needed. This is actually a too long time, and after 24 h only 20% of the chlorpyrifos was degraded. The authors should comment on results of photocatalysis with respect to literature data for this or a similar pesticide.

It is important to mention that pesticide concentration is one of the factors influencing degradation by nanocatalysts. In that sense, we are using a high concentration of chlorpyrifos (50 mg. L-1).  At higher chlorpyrifos concentrations, the production of hydroxyl radicals (OH) decreases due to a lack of active sites. Therefore, it is reasonable to expect that degradation will initiate slowly (20% in the first 24 hours) and increase over time.

The following text was added to the discussion of the revised manuscript:

“It has been reported that in photocatalytic assays as the pesticide concentration increases, the degradation rates decrease. This can be attributed to pesticides binding to active sites on nanocatalysts, as well as a reduction in UV light penetration into the solution. Consequently, both effects lead to a reduction in reactive oxygen species (ROS) production [50]. Therefore, it can explain the differences between our results and those obtained by Fadei and Kargar, who used 6 mg.L-1 CP in the assays.” (Lines 359-365)

  • Have the authors checked the nature of the composites after application to photodegrade the pesticide, i.e. after the 120 h?  Have been the same composite samples applied for more photodegradation tests?

Figure 11 of the revised manuscript shows FTIR analysis of nanocomposites before and after six cycles of photocatalytic treatment of CP revealing no discernible alterations.

  • Please define TTIP at line 113, page 4.

TTIP was defined as Titanium isopropoxide in the revised manuscript (Line 116)

Reviewer 2 Report

Comments and Suggestions for Authors

The current manuscript entitled “Catalytic role of superparamagnetic iron oxide nanoparticles as support for TiO2 and ZnO on the chlorpyrifos photodegradation in aqueous solution” deals with the synthesis, characterization, and photocatalytic properties of Fe3O4-based heterostructures. The manuscript is well-written and data analysis for the hybrid mixtures is justified. However, several points must be improved before being considered for publication in Nanomaterials.

Additional comments.

1. Corresponding authors should be specified in the manuscript.

2. The full name of TTIP should be provided in the manuscript.

3. Advanced oxidation process (AOP) is a leading technique for pollutant degradation under UV-visible light irradiation. The current manuscript should be updated with the recent references related to SiO2, TiO2, and ZnO-based photocatalysts (Molecules 202328, 1886. https://doi.org/10.3390/molecules28041886).

4. FT-IR data or XRD data of pure TiO2, SiO2, and ZnO should be compared in the text to differentiate the formation of the heterostructures or only physical mixtures. 

5. The authors should provide and discuss in the text the role of permanent porosity in photodegradation performances. BET surface area should be measured by using N2 adsorption isotherms at 77 K.

6. The stability of photocatalysts could be confirmed by the XRD, TEM, or FT-IR of used samples after several chlorpyrifos degradation progress.

7. The radical trapping experiments should be provided to confirm the role and nature of reactive oxygen species involved in the photodegradation of chlorpyrifos.

8. Why does SPION@SiO2@TiO2 show a higher photodegradation rate compared to SPION@SiO2@ZnO?

9. Transient fluorescence lifetime was the important factor for photocatalytic chlorpyrifos degradation, and the relaxation lifetime of photo-excited electron–hole pairs should be provided in the manuscript. 

10. Chlorpyrifos is not very familiar with photodegradation substrate. Its chemical structure should be depicted in the manuscript. The photocatalytic degradation mechanism of chlorpyrifos should be shown in the diagram images. 

11. There are so many new photocatalysts for environmental applications that adequate benchmarking is needed to assess at least a minimum interest. The authors should provide pieces of evidence for the advantage of their materials. How do these results (pseudo first-order rate constants) compare with literature using other photocatalysts? 

12. How did the authors reach the ideal substrate/catalyst ratio? What is the influence of light intensity? Or even the influence of temperature? Or the amount of semiconducting materials?

13. The value of total organic carbon (TOC) of chlorpyrifos mineralization should be provided as evidence for the degradation.

14. Reference format should follow the journal’s standard. Authors should correct all the references as follows. 

Author 1, A.B.; Author 2, C.D. Title of the article. Abbreviated Journal Name YearVolume, page range.

Author Response

We are thankful for the reviewers’ valuable comments and contributions to the manuscript. We hope that the present improved version of the manuscript is suitable for publication in Nanomaterials.

  • Corresponding authors should be specified in the manuscript.

The corresponding authors were specified in the revised version.

  • The full name of TTIP should be provided in the manuscript.

TTIP was defined as Titanium isopropoxide in the revised manuscript (Line 114)

  • Advanced oxidation process (AOP) is a leading technique for pollutant degradation under UV-visible light irradiation. The current manuscript should be updated with the recent references related to SiO2, TiO2, and ZnO-based photocatalysts (Molecules202328, 1886. https://doi.org/10.3390/molecules28041886).

The reference was included in the revised version (Line 61).

  • FT-IR data or XRD data of pure TiO2, SiO2, and ZnO should be compared in the text to differentiate the formation of the heterostructures or only physical mixtures. 

We conducted repeated FTIR and XRD analyses for both nanocomposites but could not definitively confirm the absence of a heterostructure due to the absence of peak displacement. Regarding the SPION@SiO2@TiO2 samples, the interference of the Fe3O4 signal makes it challenging to accurately determine any shift in the peaks. Despite efforts to mitigate noise, no discernible shift in peaks compared to ZnO was observed. However, this analysis does not conclusively establish the presence of a heterostructure. To further investigate, small-angle X-ray Scattering (SAXS) would be preferable, but unfortunately, our current XRD equipment does not support this. We acknowledge the need for this analysis in future studies. Nonetheless, TEM images present evidence countering the formation of a core-shell structure. (Lines 200-205).

  • The authors should provide and discuss in the text the role of permanent porosity in photodegradation performances.   should be measured by using N2 adsorption isotherms at 77 K.

In the revised manuscript, the Brunauer–Emmett–Teller (BET) theory was employed to calculate the specific surface areas. Additionally, the pore size distributions and pore volumes were determined using desorption data derived from adsorption–desorption isotherms, utilizing the Barrett–Joyner–Halenda (BJH) theory. The comprehensive results are presented in Table 1. (Line 270).

  • The stability of photocatalysts could be confirmed by the XRD, TEM, or FT-IR of used samples after several chlorpyrifos degradation progress.

It is a good suggestion. However, due to the limited time available to prepare a revised manuscript, we were only able to conduct FTIR analyses before and after six cycles of use. These results were illustrated in Figure 11, revealing no discernible changes in the nanocomposites. (Figure 11). (Lines 431-437)

  • The radical trapping experiments should be provided to confirm the role and nature of reactive oxygen species involved in the photodegradation of chlorpyrifos.

This is a very interesting suggestion; however, the main focus of this work was to synthesize a nanocomposite with a high potential for photocatalytic degradation of CP. In future studies, we hope to evaluate more intensively the exact mechanisms of degradation of CP. Nevertheless, existing literature has documented the collaborative involvement of •OH and O2•‾ in the degradation process for photocatalysts of similar types (Lines 389-390)

  • Why does SPION@SiO2@TiO2show a higher photodegradation rate compared to SPION@SiO2@ZnO?

In general, SPION@SiO2@ZnO and SPION@SiO2@TiO2 are very similar, however, the difference in their degradation rate stems from the fact that SPION@SiO2@TiO2 has a higher surface area which improves the ability to harvest incident light and create more surface-active sites. (Lines 390-393)

  • Transient fluorescence lifetime was the important factor for photocatalytic chlorpyrifos degradation, and the relaxation lifetime of photo-excited electron–hole pairs should be provided in the manuscript. 

This is a very interesting suggestion, we know the importance of these results, so we are working in collaboration with other teams to develop it in future studies.

  • Chlorpyrifos is not very familiar with photodegradation substrate. Its chemical structure should be depicted in the manuscript. The photocatalytic degradation mechanism of chlorpyrifos should be shown in the diagram images. 

In the revised manuscript, HPLC MS/MS (Sciex 3200 Q-trap) results were included and a degradation pathway of chlorpyrifos on SPION@SiO2@TiO2 was proposed. (Lines 439-455)

Proposed pathway of photocatalytic degradation of Chlorpyrifos by SPION@SiO2@TiO2

  • There are so many new photocatalysts for environmental applications that adequate benchmarking is needed to assess at least a minimum interest. The authors should provide pieces of evidence for the advantage of their materials. How do these results (pseudo first-order rate constants) compare with literature using other photocatalysts? 

These nanocomposites emerge as a promising alternative for the degradation of commonly used pesticides. Notably, these nanomaterials exhibit sustained magnetic and photocatalytic properties even after six cycles of use. Furthermore, their efficacy extends to the degradation of TCP, a by-product known to be more toxic than the original pesticide. While optimization is imperative to reduce degradation times.

  • How did the authors reach the ideal substrate/catalyst ratio? What is the influence of light intensity? Or even the influence of temperature? Or the amount of semiconducting materials?

In Figure 5 we determine the dose of nanocatalyst that we would require for the efficient degradation of 50 mg/L chlorpyrifos. This particular pesticide concentration was selected based on the pesticide use instructions for different crops and pests. We did not determine the influence of light intensity, we selected UV-A as it has the longest wavelength and it has been reported that the shorter the wavelength the more pesticide degradation is achieved (even without using photocatalysts). If we wanted to increase the light intensity, we would have to change the bulb or move it closer to the Erlenmeyer flasks. In both cases, we would be increasing the temperature at which the reaction occurs which is an undesirable effect. In this sense, we tried to maintain a temperature as close to room temperature as possible.

  • The value of total organic carbon (TOC) of chlorpyrifos mineralization should be provided as evidence for the degradation.

We do not consider the TOC study because the results indicate that complete mineralization of chlorpyrifos does not occur; instead, it only degrades to TCP, and TCP degradation reaches 50% after 120 hours. Mineralization involves the complete degradation of the pesticide and its by-products to CO2 and H2O, which has not been observed. We believe that this analysis could be performed in future studies if we achieve a higher rate of degradation of TCP, as well as with HPLC ms/ms to identify possible by-products generated during degradation.

  1. Reference format should follow the journal’s standard. Authors should correct all the references as follows. 

Author 1, A.B.; Author 2, C.D. Title of the article. Abbreviated Journal Name YearVolume, page range.

The reference format was corrected in the revised version of manuscript.

Reviewer 3 Report

Comments and Suggestions for Authors The research results on the catalytic role of ultra-small iron oxide
nanoparticles in TiO2 and ZnO nanocomposites are promising for
applications in the photodegradation of Chlorpyrifos-type pesticides,
especially since it is possible at pH 7. The concept presented by
the authors seems to be correct. I would just like to point out that
the method could probably be extended toward the use of mesoporous
silica with nanoparticles synthesized in the pores of mesoporous
silica. The results in the manuscript could be enriched with measurements
related to the agglomeration effect of the nanocomposites.
Nevertheless, the current version of the manuscript is consistent.
I suggest to publish it in its present form.

Author Response

We appreciate your comments, the mesoporous structure of the silica in the nanoparticles was emphasized and discussed in the revised manuscript. Additionally, we have incorporated various enhancements that address and expand upon the points raised during the review process. We believe that these modifications contribute significantly to the overall clarity of the manuscript.

Round 2

Reviewer 2 Report

Comments and Suggestions for Authors

The revised manuscript is now acceptable for publication.